# Research on the Application of Synthetic Polymer Materials in Contemporary Public Art

**DOI:** 10.3390/polym14061208

**Published:** 2022-03-17

**Authors:** Yajun Zhang, Xuelian Yu, Zhichao Cheng

**Affiliations:** 1Faculty of Design and Architecture, University Putra Malaysia, Serdang 43400, Selangor, Malaysia; 10081521@aufe.edu.cn; 2School of Art, Anhui University of Finance and Economics, Bengbu 233000, China; 20081521@aufe.edu.cn; 3Graduate School, University of Perpetual Help System DALTA, Manila 99005, Philippines

**Keywords:** synthetic polymer materials, public art, application, multidisciplinary

## Abstract

Synthetic polymer materials are widely used in contemporary public art creation. This review summarizes the application methods and current situation of synthetic polymer materials in public art, analyzes the reasons behind them and points out the deficiencies in this research field. Finally, the development trend of the interaction between synthetic polymer materials and public art is put forward.

## 1. Introduction

A world without plastics, or synthetic organic polymers, seems unimaginable today [1]. As with metal materials and ceramic materials, polymer materials are an important material type widely used in current human social activities and are one of the cornerstones for building a good life in modern society [2]. Usually, based on the source, polymers can be classified as natural polymers and synthetic polymers, but also as natural polymers, semi-synthetic polymers and synthetic polymers [3]. Polymer materials are widely used in production and life because of their easily accessible source, processing and excellent performance [4]. Synthetic polymer materials refer to polymers obtained by certain polymerization reactions using monomers with a known structure and relative molecular weight as raw materials, mainly starting from phenolic resin. There are many types of synthetic polymer materials, including plastics, synthetic fibers and synthetic rubber, which are the products of the development of modern materials science and are widely used in all aspects of production and life. The industrial production of phenolic resin in 1907 marked the beginning of human application of synthetic polymer materials. Since then, the types of polymer materials synthesized and industrially produced have developed rapidly. In the late 1960s, new products emerged one after another, synthesizing plastic materials with various characteristics, such as polyoxymethylene, polyurethane, polycarbonate, polysulfone, polyimide and polyetheretherketone. Special coatings, adhesives, liquid rubber, thermoplastic elastomers and high-temperature-resistant special organic fibers have been synthesized, making synthetic polymer materials indispensable materials for national economies and daily life [4,5].

In the development of human history, the history and development of public art and materials have gone hand in hand [6]. Art is traditionally exhibited in galleries, and buildings show exclusivity. In contrast, art nowadays is scattered around outdoor public spaces for ordinary people to enjoy. This type of art is called public art, which is art for the public. It is the physical expression of ideas, feelings and messages to public viewers in a public space. Public art includes not only public facilities with decorative functions but also permanent or temporary artwork set in a public space, such as sculptures, murals, reliefs and installations. Public art can not only beautify people’s living space and improve people’s living standards but can also help to cultivate a regional cultural temperament and clarify a nation’s cultural identity, which has important cultural value [7,8]. Since the 1960s, public art has been paid increasing attention and has played a role in beautifying the environment in the urban construction of Western developed countries. After the 1970s, with the transfer of global manufacturing from developed countries to developing countries, the wave of urbanization rose all over the world. During this period, public art was employed as a means to revitalize national economies, reflecting its important economic value [9]. Public art is also an important material medium for governments to establish a good relationship with the public. People can feel the progress and changes of the city through the public artwork in the urban space and can also express their demands, which plays a promoting role in alleviating social contradictions and enhancing social cohesion, which embodies its important political value [10].

From the interdisciplinary perspective, this paper describes the present situation of the application of synthetic polymer materials in contemporary public art, explores the application methods of synthetic polymers in contemporary public art, analyzes the deep-seated reasons for the wide application of synthetic polymer materials in public art and points out the shortcomings of current research in this field. Lastly, the cross-development trend between synthetic polymer materials and public art is put forward. Through the research methods of literature research, data collection and case analysis, this study discusses the facts, processes, mechanisms and future possibilities of the integrated development of synthetic polymer materials and public art, in order to provide new ideas and possibilities for the further innovation and coordinated development of public art and polymer materials.

## 2. Research Background

In order to effectively obtain research literature on the application of polymer materials in public art, as of January 2022, the authors first searched the Chinese literature database CNKI with the term “public art, materials” through “theme”. CNKI returned 151 related documents. Among them, there were 93 academic journals, 30 master’s theses, 1 doctoral thesis and 1 conference thesis. The earliest literature was published in 2002. The main topic data are as follows (Table 1).

After a systematic study, the discussion of “materials” in the Chinese literature mainly focuses on the color, texture and cultural connotation of public art materials [11,12,13]. The literature concerning “new materials” tends to discuss the application of new media technologies such as sound, light and images in public art [14,15,16], or from the perspective of environmental protection or recycled materials [17,18]. No research results on the theme of synthetic polymer materials were found in the obtained literature. A small number of studies briefly introduced polymer materials as a material in public art from the perspective of the art discipline [15,19,20].

In the SCOUPS literature database, the string “public art” and “material *” was searched, with irrelevant studies eliminated, and a total of 157 related studies were obtained. The main types of studies were as follows: 85 academic papers, 37 conference papers, 17 monographs and chapters, 11 reviews and 3 works. The earliest literature was published in 1994. The main subject data are shown in the following table (Table 2).

Through a systematic study, the discussion of public art materials in the English literature mainly focuses on the sociality of public art [21,22], and public [23,24], cultural [25] or narrative analysis of materials in specific cases [26,27]. Although some studies presented discussions from the perspective of engineering and materials science, some focused on ceramic materials [28], and some on the preservation and restoration of public art [29,30]. There was a paper discussing materials science research from an interdisciplinary perspective, but the theme of the discussion was art education [31].

The above literature provides a theoretical basis for the analysis of specific materials in this paper, and at the same time, the following conclusions can be drawn: First, as a relatively young art form, the research on public art started recently within the last twenty years. Secondly, the research on contemporary public art mainly focuses on the perspective of sociology and art, while the research on contemporary public art from the perspective of materials science is still relatively weak. Finally, although there are many studies on polymer materials and environmental art and product design, the research on synthetic polymer materials and public art has not yet been initiated.

## 3. Application Forms of Synthetic Polymer Materials in Contemporary Public Art

### 3.1. Requirements for Physical Properties of Public Art Materials

In order to illustrate the importance of synthetic polymer materials in contemporary public art, the performance requirements of public art materials are briefly introduced. The basic performance of materials directly affects the selection of materials in public art creation. Designers choose materials according to the presentation effect of works of art and different conditions such as the environmental temperature and climate. Please see the physical properties that have a great influence on public art in Table 3.

### 3.2. Application Forms of Synthetic Polymer Materials in Contemporary Public Art

Materials have one of the most important roles that can complete artistic creation. Any art that cannot be presented through materials is just a castle in the air [33]. The development of public art is limited by the physical properties of materials themselves, but synthetic polymer materials effectively make up for the shortage of materials and broaden the selection range of materials in public art.

Traditional public sculpture works often use a single material or make the shape as a whole in the sculpture design, so as to avoid a fracture of the connection point and other problems. To some extent, this limits the richness of modeling and materials of public sculpture works. With the continuous expansion of the volume of urban space, larger spaces need larger works, but the corrosion resistance of steel cannot meet the requirements of outdoor public sculpture works. Public art must have a positive response to the times, which is reflected in the application of materials in the composite era, on the one hand, and the expression of ideas in the era, on the other hand [34]. Synthetic polymer materials are widely used in public art creation according to the four attribute characteristics of public art materials, volume, times and ideas.

(1) Synthetic polymer materials are used as adhesives to improve the richness of public art materials. Polymer adhesives have a long history of use in public buildings and art. As early as 2000 years ago, people in the Qin Dynasty of China began to use mortar made of glutinous rice paste and lime as the adhesive for the cornerstone of the Great Wall [35]. The polymers used in synthetic adhesives, in addition to curing agents, softeners, inorganic fillers and solvents, can be added according to the situation. They have many varieties, offer excellent performance and are widely used adhesive materials [36]. The synthetic polymer adhesive widely used in contemporary public art is a two-component adhesive, which is commonly known as AB adhesive; epoxy resin AB glue is formed by an epoxy resin (component A) and a polyfunctional hardener (component B). There are two-component adhesives with acrylic acid, epoxy, polyurethane and other components on the market, which are easy to purchase. The adhesive has the advantages of a high bonding strength, high hardness and high chemical resistance, and it is simple to operate, can be cured at low temperature, room temperature, medium temperature or high temperature and has strong adaptability. This type of adhesive is widely used in the installation and repair of environmental ceramic art [37] and also in the bonding of different components and decorative surfaces in public artwork. In addition, EVA (ethylene-vinyl acetate) resin hot-melt adhesive (hot glue) is often used to bond lower-weight work components. Hot-melt adhesive has been widely used because it is easy to operate, can be glued in an instant and can be operated repeatedly, which can best meet the requirements of public art creation and bonding. The most commonly used hot-melt adhesives are ethylene-vinyl acetate copolymer resin, ethylene-acrylic acid copolymer resin, polyphthalamide resin, polycool resin, etc. For example, in 2021, Zhu Lin and Mumu produced “Flowers of Velvet, Mirror Image of Spring” (Figure 1) for the Chinese Cultural Center in the Winter Olympic Village of the Beijing Winter Olympics, which used a large number of hot-melt adhesives [38].

(2) Synthetic polymer materials are used as external coatings to improve the weather resistance of materials for large-scale public works of art. Because public art is often set in an outdoor space, the weather resistance of materials is extremely high. Steel structures have high strength and good toughness, but their corrosion resistance is poor and they can easily rust, which seriously limits their application in public art. In 1982, the Asahi Glass Company of Japan developed fluoroolefin-vinyl ether copolymer (FEVE), which created a room-temperature curable fluororesin that can be dissolved in aromatic hydrocarbon, lipid and ketone solvents at room temperature. It overcame the disadvantage that the original fluorocarbon coating cannot be cured at room temperature, realized the ideal of coating fluorocarbon coatings on construction sites and greatly expanded the application field of fluorocarbon coatings [39]. Fluorocarbon coating is a general designation for a series of coatings with fluorine-containing resin as the main film-forming material. It is a coating material modified and processed on the basis of fluorine resin. Its main feature is that the resin contains a large number of F-C bonds, with a bond energy of 485 kJ·mol^−1^, which is the highest among all chemical bonds [40]. For example, for the public artwork “The Torch” (Figure 2), which was up to 36 m high and welded from steel, the bright red fluorocarbon coating is an important factor in the success of this work. In addition, various types of epoxy resins prepared are widely used in the anti-corrosion treatment of stone and wood, effectively improving the weather resistance of these materials and greatly broadening the material selection space for public art. Because ordinary pigments cannot bear the outdoor environment, pigments blended with synthetic polymer materials are directly used by artists to decorate public works of art, enriching the colors and means of expression of public art.

(3) Synthetic polymer materials, as the main material, embody the characteristics of the times of public artwork. Take plastics as an example. Plastics are organic synthetic materials made of resin which have certain chemical corrosion resistance and impact resistance, and there are many types, such as polyethylene terephthalate (PET or PETE), high-density polyethylene (HDPE), polyvinyl chloride (PVC or vinyl), low-density polyethylene (LDPE), polypropylene (PP) and polystyrene (PS or Styrofoam). Plastics can be seen everywhere in modern life and are widely used in public art. They are a relatively energy-saving material, easy to process and low in cost. They can keep the shape of objects unchanged at normal temperature. In many cases, plastics are also used as a substitute for many natural materials, such as plexiglass plates and artificial marble. Plastic is a light and easy-to-form material too. The famous public artwork “Rubber Duck” (Figure 3) created by Dutch artist Florentin Hoffman, one of the most famous contemporary public artworks made of plastic materials, uses plastic materials that are light and easy to form to create huge works and make them float on the water [41].

(4) Synthetic polymer materials appear in public artwork in the form of ready-made products, which expresses the public art’s attention to current social problems. In order to arouse the audience’s attention, contemporary public art often directly uses articles in daily life as raw materials for works of art. Due to the close relationship with people’s daily life, synthetic polymer materials often appear in contemporary public artwork. For example, discarded plastic products are often used in public welfare design, advocating the concept of environmental protection. As a representative of public artwork using polymer ready-made product materials, “Whale” (Figure 4) is made of five tons of plastic waste from Hawaii, which is intended to arouse people’s attention concerning marine protection [42].

## 4. The Widespread Application of Synthetic Polymer Materials in Public Art and Analysis of the Reasons

### 4.1. Synthetic Polymer Materials Have Become One of the Most Important and Common Main Materials in Public Art

Compared with traditional sculpture materials such as marble and brass, synthetic polymer materials are not suitable for making traditional conceptual public artwork. Long-term exposure to ultraviolet radiation will cause discoloration, structural strength reduction and poor resistance to changing temperatures; some materials begin to crumble after several years of exposure to weather conditions. Additionally, probably the most important aspect is the problems with recycling. What makes synthetic polymer materials so dominant in the creation of contemporary public art?

As the basis for the presentation of public art, materials’ position in public art has been paid increasing attention by artists and the public [43]. At the same time, with the progress of materials science and technology, the materials used by public artists in artistic creation have become increasingly abundant, and the role played by materials has also changed. By searching for “the best public art of 2021” on the Internet, two works will definitely be selected. One is “L ‘Arc de Triomphe, wrapped”. Christo and Jeanne-Claude’s preposterous work, which wrapped the iconic French arch in 25,000 square meters of silly blue recyclable polypropylene fabric and 3000 m of red rope, was probably 2021’s most important and visible piece of public art. It attracted worldwide media attention, with a reported six million viewers on the ground in Paris [44]. The other work is “In America”, which was created by Suzanne Brennan Firstenberg. In the throes of the COVID-19 Delta variant outbreak in the United States, within a country struggling with the weight of an unfathomable collective grief and battling through pernicious anti-public health propaganda, artist Suzanne Brennan Firstenberg installed her monumental In America project on the National Mall in Washington, D.C. Firstenberg and a team of associates planted small, palm-sized polyester fiber white flags en masse, each representing a life lost to the COVID-19 pandemic. The flags were placed on thin wires, installed by piercing the ground and, in turn, blanketed the north lawn around the Washington Monument [45].

### 4.2. Reasons for the Rapid and Widespread Application of Synthetic Polymer Materials in Public Art

#### 4.2.1. Synthetic Polymer Materials Fit the Core Value of Contemporary Public Art

Contrasted with other art forms, the most important feature of public art lies in its publicity. It can be said that publicity is the core of public art values. Sociologically speaking, “public” in “public art” refers to a social field, specifically, a public field [46]. The striking feature of contemporary public art involves the extensive participation of the public. It directly targets and serves the public, showing the characteristics of openness, kindness and freedom. Traditional public spaces are often well-planned squares with magnificent memorial sculptures in the middle, which can only be used to hold solemn and important activities. Contemporary public spaces are created through the intervention of public art, allowing these venues to become places for public communication and interaction. The public can feel the artistic body language and the affinity of art through the open and interesting artistic forms of public art.

Contemporary public art plays a vital role in the change and exploration of materials in the process of dispelling the solemn and dignified form and core value of traditional urban sculptures. Since Duchamp sent urinals to an exhibition in 1917, the boundaries between high and low materials in artistic works have completely disappeared, breaking the sense of distance and alienation between traditional shelf art and the audience [14]. Synthetic polymer materials have spread all over people’s lives today. They naturally have the spiritual and cultural demands of the core value of contemporary public art. They are the most distinguishable material from traditional public sculpture materials such as metal, stone and ceramics, and they are also the most representative material far away from “elite” esthetics. At the same time, they are also the type of material that the public is most familiar with, most likely to have affinity with and most likely to accept. Therefore, because of their absolute “dominance” over people’s daily life, synthetic polymer materials have naturally and rapidly become materials frequently used by contemporary public artists, which is the cultural and social reason why synthetic polymer materials are widely used in contemporary public art.

#### 4.2.2. The Excellent Performance of Synthetic Polymer Materials Meets the Practical Needs of the Development of Contemporary Public Art

(1) Compared with other artistic materials, polymer materials have obvious advantages in physical properties. First of all, synthetic polymer materials have high strength and good toughness, which cannot be compared with ceramics, glass, wood and other materials. This ensures that polymer materials have strong anti-wear ability in application, which is suitable for public works of art such as stations and shopping malls with large flows of personnel and more contact with the public. Secondly, polymer materials have strong corrosion resistance. In rainy and snowy weather or places with harsh climatic conditions, their chemical properties are stable, and problems do not easily occur. Compared with steel or other organic materials, synthetic polymer materials are more suitable for works in seaside and outdoor spaces. Thirdly, polymer materials are lighter than other traditional public art materials. The lighter mass makes it possible for synthetic polymer materials to form works with more difficult structural mechanics, thus presenting a more dramatic artistic image. Fourthly, synthetic polymer materials have the advantages of easy processing and modification [47]. Polymer materials are rich in molding and processing methods, such as laser molding, calendering, injection molding, blow molding and extrusion molding. These molding methods can realize public works of art with various shapes. Finally, synthetic polymer materials cover a wide range. All types of plastics, synthetic rubber, synthetic fibers, synthetic resins, etc., can fully meet the needs of various engineering technologies such as structural support, building attachment, inflation, wrapping and appearance filling in contemporary public art.

(2) Synthetic polymer materials can reduce the production cost and shorten the production cycle of public works of art. The increasingly common setting of public art and public facilities requires lower production costs and more efficient process methods. In the actual creation of public art, the materials and processing costs often occupy a large part of a public art project’s entrustment fees, especially for works which are placed in commercial spaces (Table 4). Relatively low prices and relatively short processing and construction times have become problems that public artists have to consider when choosing materials. The authors investigated the sculpture materials commonly used in public art through the Internet and found that synthetic polymer materials have obvious advantages in cost and production cycle compared with traditional sculpture materials such as metal, wood and stone (Table 5).

(3) In public art, synthetic polymer materials have richer and freer forms of expression. Artistic creation cannot be separated from the support of materials, and the richness of artistic materials can provide more possibilities for artistic creation. The color of public spaces has a strong visual impact on people, and the color of polymer materials is extremely rich. For example, the melamine resin coating material has already proved its value in public art design because of its infinite design potential. Some polymer materials, such as plastics, can show almost any desired color after adding various fillers and colorants during processing.

In addition to color, synthetic polymer materials can achieve a varied texture in public art. The texture of a material is the reflection of the human sensory system’s response to the material due to physiological stimulation or the information obtained from the surface characteristics of the material by the human perceptual system [48]. The material is reflected by both vision and touch, and the esthetic feeling given to people includes the pleasure conveyed by touching. Synthetic polymer materials are good at imitating wood, stone, steel or other expensive materials because of their good controllability. Early polymer materials brought a reputation of being low-level and cheap because of their unreal personality. Through continuous development, polymer materials can not only confuse the real with the fake but also create their own unique personality. They can realize various changes such as in roughness, smoothness, softness and hardness, cold and warm, luster and transparency and elasticity. They seem to be omnipotent. This is undoubtedly very important for public art that emphasizes affinity and zero distance with the public.

Synthetic polymer materials can also break through the traditional modeling concept and create new esthetic feelings and forms. For example, high-strength thin-film materials in synthetic polymer materials have laid the foundation for the wide application of membrane structures. Because of their reasonable stress, beautiful shape and economic advantages, they are widely applied in vibrant modern public buildings. Because of their light weight, membrane buildings can cover space with a large span without internal support. The flexibility of materials allows people to design and use space more flexibly and creatively. For example, Saudi Arabia’s International Terminal uses a hyperbolic membrane roof supported by a mast to ensure a long-span tent structure with fiber tension under wind load in any direction, creating a unique soft curve in the internal space, which is concise and lively and conforms to the characteristics of the times. The Ulubo Church designed by designer Jay Cha determines the best pattern of light and shadow through a translucent polycarbonate vinegar plate as the sun moves around the building. Optical fibers are applied to create a short-term feeling for the space. Different layering technologies are adopted, and the opening and closing methods of panels can be changed to produce frame scenes. The level of indoor light is also adjusted with the position of the structure in the site, making the building a sculpture of light using light translucent panels [49].

(4) The continuous development of synthetic polymer materials has continuously improved their disadvantages in the application of public art. Although synthetic polymer materials have many advantages, they also have some disadvantages such as their quick aging and difficult degradation, which restrict their application in public art to some extent. However, with the continuous progress of polymer material technology, these shortcomings are being effectively improved. If a plasticizer is added in the molding process of the material, the machinability of the material can be improved, and the glass transition temperature can be reduced to improve the cold resistance of the material. The addition of amine antioxidants, phenolic antioxidants, sulfur-containing organic compounds and phosphorus-containing compounds in the processing of synthetic polymer materials can quickly react with peroxide radicals, thus effectively slowing down the oxidation process of materials. In the manufacturing process of materials, if light stabilizers such as light shielding agents, ultraviolet absorbers, quenchers and free radical scavengers are added, the photoaging degradation of materials can be avoided [50]. Through anti-aging treatment, the weather resistance of synthetic polymer materials has been greatly improved, and the door has also been opened for their wide application in public art.

(5) Composite technology effectively enhances the properties of synthetic polymer materials. Composite polymer materials are multiphase solid materials formed by the composite bonding of polymer materials and other substances with different compositions, shapes and properties, and they also have interfaces [51]. The greatest advantage of composite polymer materials is that they can combine the advantages of various materials, such as high strength, light weight, temperature resistance, corrosion resistance and thermal insulation. In the 21st century, more than 90% of the composites used in production are polymer matrix composites. For example, glass fiber-reinforced plastic is the most widely used composite polymer material in public art. It takes unsaturated polyester resin, epoxy resin and vinyl ester resin as the main raw materials and becomes a transparent solid after adding a curing agent, which looks like glass, hence why it is called glass fiber-reinforced plastic. In order to prevent the liquid resin from flowing rapidly during curing, a certain proportion of talcum powder or gypsum powder is added, and synthetic fibers are added to increase the strength of the material after molding. Glass fiber-reinforced plastic is light but strong, with high strength and anti-corrosion performance. The resin is colorless and transparent, which can not only be made into a glassy body with a certain transparency but also be processed into colored glass fiber-reinforced plastic by adding various pigments [52].

The development of degradable polymer materials makes synthetic polymer materials more environmentally friendly. They are difficult to be degraded naturally, which is one of the reasons why they have been criticized in the application of public art. For example, polyethylene, polypropylene and polyvinyl chloride all take hundreds of years to degrade, and waste plastics are usually treated through landfill and incineration, resulting in increasingly serious environmental pollution problems [53]. Degradable plastics refer to plastics whose properties can meet the use requirements and remain unchanged during the shelf life, but can be degraded into environmentally friendly substances under natural environment conditions after use [54]. According to the source of raw materials, they are mainly divided into three categories: petroleum-based, bio-based and coal-based. Petroleum-based degradable plastics include polybutylene terephthalate/adipate (PBAT), polybutylene succinate (PBS) and polycaprolactone (PCL). Bio-based degradable plastics include starch-based plastics, cellulose plastics, polylactic acid (PLA) and polyhydroxyfatty acid (PHA). Coal-based degradable plastics include polyglycolic acid (PGA) [55]. At present, starch-based polylactic acid has been widely used in industrial production and daily life. It is believed that public artwork made of degradable polymer materials such as polylactic acid will soon become popular.

## 5. The Insufficiency of the Application of Synthetic Polymer Materials in Contemporary Public Art

Although synthetic polymer materials play a vital role in the development of public art, there are still many shortcomings in the application of synthetic polymer materials in contemporary public art.

First of all, the theoretical research on the application of synthetic polymer materials in contemporary public art is insufficient. This can be seen from combing through the previous research results. Synthetic polymer materials have become unavoidable materials in public art creation and have played an important role in promoting the development of public art. Under this background, the lack of corresponding theoretical research should not exist. Theoretical research can play a key summing-up, guiding and leading role in practical creation. The lag or even vacancy of theoretical research reflects the lack of rational understanding and consciousness of the application of polymer materials in public art in academic circles, and it is still in a spontaneous unconscious state. Academic circles lack the necessary discussion on the facts, processes, mechanisms and future possibilities of the combined development of these two fields, which is not conducive to the further innovation and integration of public art and polymer materials.

Secondly, there is a lack of interdisciplinary exploration and cooperation, which is reflected in both theory and practice. In terms of theory, it has been mentioned earlier that the research from the multidisciplinary perspective is relatively scarce, and in-depth theoretical exploration jointly carried out by scholars of materials science and art science is even more difficult to find. In practice, due to the lack of support from materials disciplines and scholars, the application of synthetic polymer materials to public works of art is generally at a relatively low-end civil level. Most of the polymer materials used are products that can be purchased anywhere in the market. These products have a low technical content and are easy to operate. Although these common civil materials have balanced properties, they sometimes cannot meet the requirements of a specific performance aspect in public art. Due to the lack of professional materials science support, many excellent public artwork design schemes can only stay at the stage of design drawings. Moreover, due to the lack of knowledge of polymer materials, artists’ imagination and creativity are also limited. At the same time, due to the lack of artists’ “strange” material property requirements for polymer materials science, many opportunities for polymer material innovation have also been missed.

## 6. Conclusions and Prospects

Synthetic polymer materials have become some of the most important materials in contemporary public art creation. There is strong evidence that many influential contemporary public artworks use synthetic polymer materials. Synthetic polymer materials are mainly used in public art in the following four ways. First, they are used as a strong adhesive to bond different parts of the work. Second, they are used as the external anti-corrosion and color coatings of works, which not only greatly improve the tolerance of public artwork to the harsh environment but also provide rich colors. Third, they are used as the main material of the work, sometimes even the only material. Fourth, they have become the constituent materials of public works of art in the form of ready-made products. The reasons why synthetic polymer materials are so widely used in contemporary public art are mainly the following two points: First, synthetic polymer materials are the most common and hackneyed materials in people’s daily life. This conforms to the value concept of contemporary public art, which is to try to eliminate the distance between artwork and audiences and emphasize the affinity of people. Second, the excellent performance of synthetic polymer materials relative to traditional sculpture materials meets the practical needs of the development of contemporary public art. These excellent characteristics include: more stable physical properties, lower price, shorter production cycle, more novel and free expression forms and richer innovation possibilities. In addition, the rapid improvement in the performance of synthetic polymer materials in environmental friendliness, weather resistance and degradability has made up for the shortcomings of the materials to a certain extent, making them more popular in the field of public art. However, compared with the fruitful practical achievements, the theoretical research on the application of synthetic polymer materials in contemporary public art is still relatively weak. There is still much room for further cooperation between experts in polymer materials science and art. Although this study broadly describes and analyzes the application of synthetic polymer materials in contemporary public art creation from a macro perspective, there are still deficiencies in the discussion of the application possibility of different synthetic polymer materials in public art creation. It is hoped that this can be deeply discussed in the follow-up research.

Science and art are inseparable, just like two sides of a coin [56]. Contemporary public art and synthetic polymer materials have formed an inseparable development trend. We have reasons to believe that the rich and changeable appearance and excellent physical properties of synthetic polymer materials will become a force beyond cultural factors such as style and genre and have a great impact on public art. In this process, experts in the field of contemporary public art who can deeply study synthetic polymer materials will be able to seize the initiative of artistic innovation. Artists who lack in-depth research on synthetic polymer materials will be limited in artistic creativity and miss more possibilities, so they will be at a disadvantage in the fierce competition. At the same time, the demand of public art for the special properties of materials will also bring more inspiration to the innovation of synthetic polymer materials. Practical application in public art will also play an important role in promoting and popularizing new polymer materials. Traditional public art creation often follows the conceptual logic of “cultural connotation, modeling and color”. Public art creation based on the properties of synthetic polymer materials is likely to become a new form of creation.

## Figures and Tables

**Figure 1 polymers-14-01208-f001:**
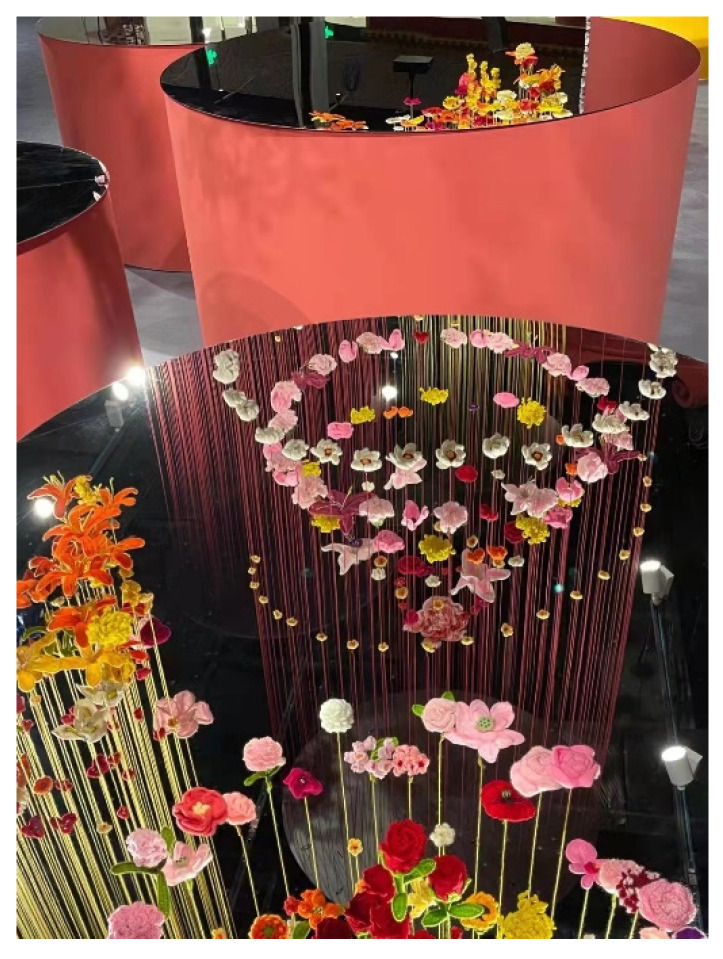
Public artwork “Flowers of Velvet, Mirror Image of Spring” [38].

**Figure 2 polymers-14-01208-f002:**
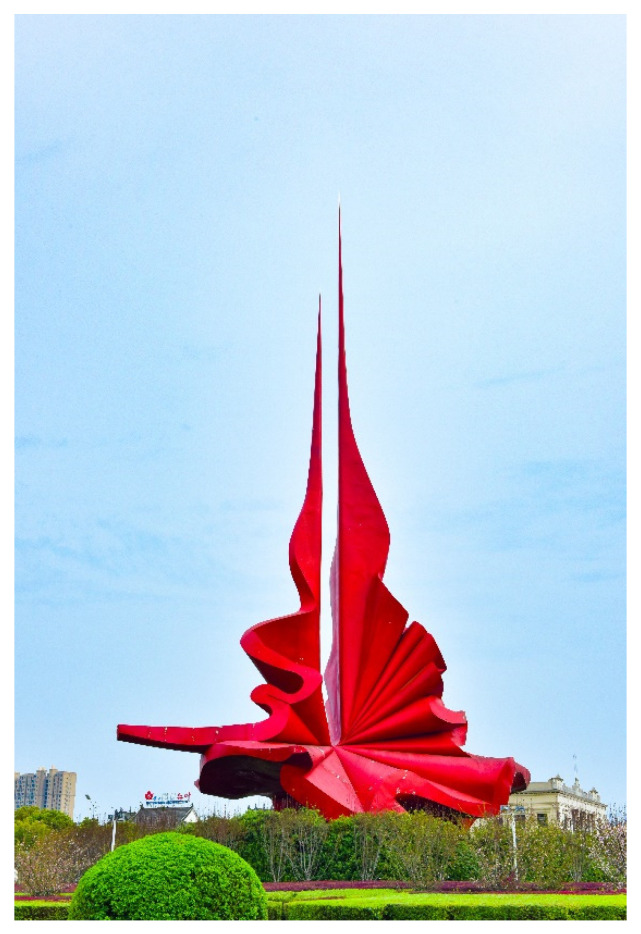
Public artwork “The Torch”.

**Figure 3 polymers-14-01208-f003:**
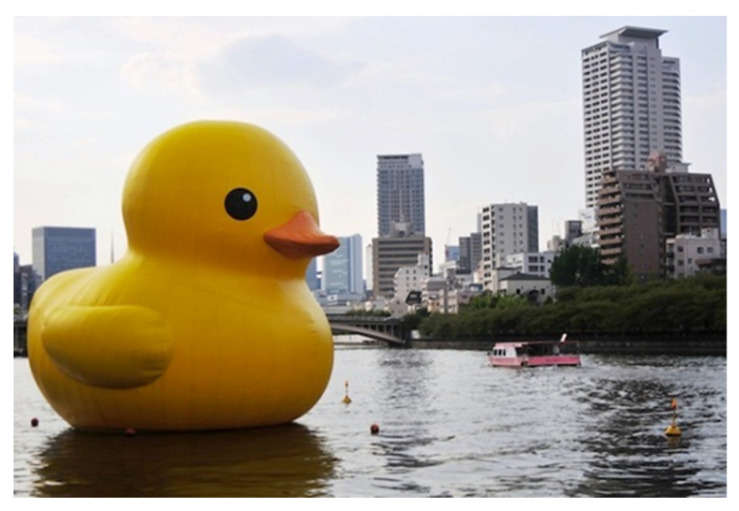
Public artwork “Rubber Duck”.

**Figure 4 polymers-14-01208-f004:**
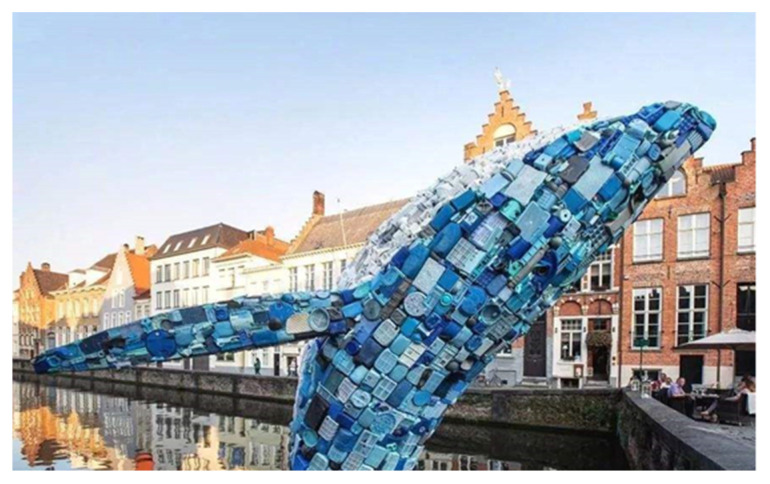
Public artwork “Whale”.

**Table 1 polymers-14-01208-t001:** Statistics of CNKI on the public art and materials literature discipline.

Public art	32	Urban public art	3
Ceramic material	11	Installation art	3
Material language	7	Esthetics	2
Subway public art	7	Material application	2
Complex material	5	Public art materials	2
Public art design	4	Low-carbon material	2
Public art major	4	Recycled material	2

**Table 2 polymers-14-01208-t002:** SCOUPS subject statistics on public art and materials-related literature.

Social Sciences	68	Earth and Planetary Sciences	3
Arts and Humanities	62	Agricultural and Biological Sciences	2
Engineering	42	Chemical Engineering	2
Computer Science	20	Chemistry	2
Materials Science	8	Economics, Economics and Finance	2
Materials Science	8	Medicine	2
Business, Management and Accounting	7	Psychology	2
Environmental Science	7	Biochemistry, Genetics and Molecular Biology	1
Mathematics	4	Decision Sciences	1
Physics and Astronomy	4	Multidisciplinary	1

**Table 3 polymers-14-01208-t003:** Requirements for physical properties of public art materials [32].

Physical Properties	Influencing Factors	Select by
Density	This will affect the hardness, strength and water absorption of the work itself. Objects with a high density are hard but difficult to process. Objects with a low density have strong water absorption and are prone to leakage.	Mechanical structure, climatic humidity
Water absorption	This will affect the durability of the work. Most materials with strong water absorption are easy to process, but they are easy to deform and decay. When placed outdoors, they must be waterproof.	Climate, location of furnishings
Abrasion resistance	Frequent human touch and sandstorms will change the color, texture and even shape of the works.	Geography and climate, affinity with audience
Corrosion resistance	In coastal areas, areas with frequent acid rain and areas around chemical plants, it is easy to be affected by acidic or alkaline substances in the atmosphere and rainwater.	Geographical climate, air quality
Shrinkage rate	This is affected by the ambient temperature and humidity, high-temperature expansion, low-temperature contraction, water absorption expansion and drying contraction. When materials with too large a difference in shrinkage are matched, deformation and cracking will occur, affecting the shape and life of the work.	Structure and material combination of works

**Table 4 polymers-14-01208-t004:** Price comparison of some polymer materials and traditional sculpture materials.

Classification	Material	Unit	Price/Yuan	Data Sources 11 February 2022
Synthetic Polymer Materials	Glass Fiber-Reinforced Plastics (FRP)	1 m^2^	90	https://b2b.baidu.com
	LCP Plastic	1 kg	43	https://b2b.baidu.com
	Thermoplastic Styrene Butadiene Rubber (SBS)	1 kg	14	https://b2b.baidu.com
	Polypropylene Fiber (PPF)	1 kg	22	https://b2b.baidu.com
	Ceramic Particle Adhesive	1 kg	21	https://b2b.baidu.com
	Hybrid Polymer Multilayer Composites	1 kg	9	https://b2b.baidu.com
Traditional Sculpture Materials	Glass	1 m^2^	160	https://b2b.baidu.com
	Metal (Copper)	1 kg	110	https://b2b.baidu.com
	Stainless Steel	1 m^2^	125	https://b2b.baidu.com
	Granite	1 m^2^	70	https://b2b.baidu.com
	Wood (Mahogany)	1 m^3^	13,000	https://b2b.baidu.com

**Table 5 polymers-14-01208-t005:** Comparison of production price, production cycle and weight of different materials for sculptures of the same size (2.4 m × 1.1 m × 1.9 m).

Classification	Material	Price/CNY	Production Cycle/Days	Weight/Ton	Data Sources 11 February 2022
Synthetic Polymer Materials	Glass Fiber-Reinforced Plastics (FRP)	7500	10–15	0.05	https://www.baidu.com https://b2b.baidu.com https://uland.taobao.com
	LCP Plastic	402.04	20–25	0.02	https://b2b.baidu.com https://uland.taobao.com
	Thermoplastic Styrene Butadiene Rubber (SBS)	416	15–20	0.032	https://www.baidu.com https://b2b.baidu.com https://uland.taobao.com
	Polypropylene Fiber (PPF)	540	20–25	0.014	https://www.baidu.com https://b2b.baidu.com https://uland.taobao.com
	Ceramic Particle Adhesive	736	15–20	0.046	https://www.baidu.com https://b2b.baidu.com https://uland.taobao.com
	Hybrid Polymer Multilayer Composites	620	10–15	0.062	https://www.baidu.com https://b2b.baidu.com https://uland.taobao.com
Traditional Sculpture Materials	Fused Glass	20,252.2	45–50	10.58	https://b2b.baidu.com https://uland.taobao.com
	Metal (Copper)	62,400	45–50	12.8	https://b2b.baidu.com https://uland.taobao.com
	Stainless Steel	24,500	30–35	9.8	https://b2b.baidu.com https://uland.taobao.com
	Granite	32,850	35–40	4.9	https://b2b.baidu.com https://uland.taobao.com
	Wood (Mahogany)	10,530	50–55	2.7	https://b2b.baidu.com https://uland.taobao.com

## Data Availability

Not applicable.

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
