# Peer review of "Research on the Application of Synthetic Polymer Materials in Contemporary Public Art"

_polymers, 2022, doi:10.3390/polym14061208_

Round 1

Reviewer 1 Report

The manuscript polymers-1627505 has been improved over the previous version and therefore, I recommend the publication of this work after minor revision:

  • The caption of Figure 3 is ambiguous! There are no explanations for 3a and 3b!
  • L. 442: “in promoting the promotion”? Please revise the sentence!
  • The Conclusions are very general! The authors mentioned in Abstract that “this review summarizes (1) the current situation of synthetic polymer materials in public art, (2) analyzes the reasons why synthetic polymer materials are widely used in this field, (3) the deficiencies in application of materials based on synthetic polymers in public art, and (4) the future development directions of synthetic polymer materials in public art”, but in Conclusion there is noneexplanation related with these statements! Please try to reformulate the conclusions in concordance with the mentioned objectives from Abstract!
  • L. 446: “Anhui provincial university”? Please revise the name of the university! In addition, “university” must be written as “University”!
  • The Reference section does not respect the conditions from the Instructions for Authors! Please revise the entire section!
  • There are still problems with English expression, which should be checked and corrected by a native Englishman!

Reviewer 2 Report

The manuscript "Research on the Application of Synthetic Polymer Materials in Contemporary Public Art" presents a relevant topic and suitable for publication in this journal, however adjustments are fundamental for its acceptance and subsequent publication, authors should focus on answering all comments below :

(1) The abstract is not adequate, the authors make a clear distinction between topics, such as methods and conclusion, and this should not happen. The abstract must be clear and concise to the readers, allowing its correct reading and interpretation!!
(2) At the end of the introduction you must enter your objective and innovation of this research.
(3) The legend of all figures must be verified.
(4) The conclusion is not good enough, note that there is only one paragraph!! Rewrite it and make it more fluid and easier to read.

Round 2

Reviewer 1 Report

The manuscript polymers-1627505 has been improved over the previous version and as a first impact, I would have liked to accept this manuscript for publication, but in the end, I saw that is still a lot of work to be done on it! Unfortunately, there are some corrections that I would like to check if are done, so I consider that the manuscript can be published after minor revisions.

Please take in consideration the following:

There are no figures in the manuscript, except for Figure 1!! The authors responded to my query that „I have added explanations for Figure 3a and 3b”, but there is no Figure 3! I imagined that this is a misprinting, but then I observed that the authors deleted all the figures! A review without figures, scheme and tables it is unacceptable, especially in this field of visual art!!! The addition of representative figures related to the contemporary public art in the content of this review is all the more necessary as readers expect a strong visual impact, as soon as they read the title of the manuscript! Please try to improve this manuscript with explicit figures and schemes!!

There is a mixture of reference styles which should never appear in a manuscript!!! For example, the references from the following paragraphs must be placed in square brackets [ ], like as it is used in the whole manuscript:

  1. 61-68: „After systematic study, the discussion of "materials" involved in Chinese literature mainly focuses on the color, texture and cultural connotation of public art materials (Wu Shixin, 2005; Teng Ying, 2012; Wang Yao, 2014; Yang Yi, 2016). Literature concerning "new materials" tends to discuss the application of new media technologies such as sound, light and image in public art (Chang Jie, 2008; Huang Xingzi, 2016; Li Yating, 2020), or from the perspective of environmental protection or recycled materials (Chen Jing, 2018; Luo Weian, 2018). No research results on the theme of synthetic polymer materials were found in the obtained literature. A small number of literatures briefly introduced polymer materials as a material in public art from the perspective of art discipline (Zhang Junli, 2009; Huang Xingzi, 2016; Ma Chuanlong, 2020).”
  2. 74-80: „Through systematic study, the discussion of public art materials in English literature mainly focuses on the sociality of public art (Gabrys·J & Yusoff·K, 2012; Campos·R, 2020;), public (Ainsworth·K, 2011; Radice·M, 2018), cultural (Leimbach·T, 2018), or narrative analysis of materials in specific cases (Daugelaite·A, 2018; Campos·R, 2020). Although some literatures are discussed from the perspective of engineering and materials science, some focus on ceramic materials (Fang·M, 2020), and some on the preservation and restoration of public art (Boulton·L, 2015; Mezzadri·P, 2021). There is a paper discussing materials science research from an interdisciplinary perspective, but the theme of the discussion is art education (Arild Berg, 2016)”.

Author Response

There are no figures in the manuscript, except for Figure 1!! The authors responded to my query that “I have added explanations for Figure 3a and 3b”, but there is no Figure 3! I imagined that this is a misprinting, but then I observed that the authors deleted all the figures! A review without figures, scheme and tables it is unacceptable, especially in this field of visual art!!! The addition of representative figures related to the contemporary public art in the content of this review is all the more necessary as readers expect a strong visual impact, as soon as they read the title of the manuscript! Please try to improve this manuscript with explicit figures and schemes!!

 Answer: Thank you very much for your pertinent comments. I really learned a lot from your comments. I'm sorry for my carelessness. Last time, after the revision and reply according to your opinion, another reviewer reminded me that the legend of all figures must be verified. I encountered this situation for the first time. I didn't know how to obtain authorization, so I had to delete some pictures. But I forgot to revise my reply to you. I’m sorry again. This time I added some pictures that can be authorized.

There is a mixture of reference styles which should never appear in a manuscript!!! For example, the references from the following paragraphs must be placed in square brackets [ ], like as it is used in the whole manuscript:

  1. 61-68: „After systematic study, the discussion of "materials" involved in Chinese literature mainly focuses on the color, texture and cultural connotation of public art materials (Wu Shixin, 2005; Teng Ying, 2012; Wang Yao, 2014; Yang Yi, 2016). Literature concerning "new materials" tends to discuss the application of new media technologies such as sound, light and image in public art (Chang Jie, 2008; Huang Xingzi, 2016; Li Yating, 2020), or from the perspective of environmental protection or recycled materials (Chen Jing, 2018; Luo Weian, 2018). No research results on the theme of synthetic polymer materials were found in the obtained literature. A small number of literatures briefly introduced polymer materials as a material in public art from the perspective of art discipline (Zhang Junli, 2009; Huang Xingzi, 2016; Ma Chuanlong, 2020).”

  1. 74-80: „Through systematic study, the discussion of public art materials in English literature mainly focuses on the sociality of public art (Gabrys·J & Yusoff·K, 2012; Campos·R, 2020;), public (Ainsworth·K, 2011; Radice·M, 2018), cultural (Leimbach·T, 2018), or narrative analysis of materials in specific cases (Daugelaite·A, 2018; Campos·R, 2020). Although some literatures are discussed from the perspective of engineering and materials science, some focus on ceramic materials (Fang·M, 2020), and some on the preservation and restoration of public art (Boulton·L, 2015; Mezzadri·P, 2021). There is a paper discussing materials science research from an interdisciplinary perspective, but the theme of the discussion is art education (Arild Berg, 2016)”.

Answer: Thank you very much for your valuable comments. I have revised it according to your comments.

Reviewer 2 Report

The authors made all corrections.

Author Response

Thank you for your comments on my manuscript. Your valuable suggestions let me learn a lot in thesis writ. I am also very grateful to you for your encouragement and recognition, which is very important for my unremitting study. Thank you again.

Round 3

Reviewer 1 Report

The manuscript polymers-1627505 has been improved over the previous version and therefore, I recommend the publication of this manuscript, only after the authors will arrange all the references within the manuscript, as it is required in the Instructions for Authors, namely: "reference numbers should be placed in square brackets [ ], and placed before the punctuation; for example [1], [1–3] or [1,3] ”!

I mentioned in the revision that there is a mixture of reference styles: either [number] throughout the manuscript, or [author's name] in paragraphs L. 61-68 and L 74-80. This is not correct and is not used anywhere in writing a manuscript!!!

Please bring all the references in the text to a common form!

Moreover, if one person were interested in one of the references presented in the paragraphs mentioned above, as an example: "(Wu Shixin, 2005; Teng Ying, 2012; Wang Yao, 2014; Yang Yi, 2016)", he would not find any clear clues in your way of presenting the references, without looking for them and finding them in the References section.

So please change the presentation of the references from [author name] to [number] in paragraphs L. 61-68 and L 74-80 and add all the references presented in the mentioned paragraphs, in the References section at the end of the manuscript, before publishing the manuscript!

Author Response

Dear Reviewer:

       Thank you very much for your valuable comments. I didn't understand you correctly in last revision because I didn't read carefully enough about the Instructions for Authors. I'm sorry for that and grateful for your patient explanation to me. The format of references in the manuscript have been unified according to your comments. Thanks again.
